# Magnetic excitations in strained infinite-layer nickelate PrNiO$_2$ films

Qiang Gao [1,13], Shiyu Fan [2,13], Qisi Wang [3,4,13], Jiarui Li [5], Xiaolin Ren[1,6], Izabela Biało [3,7], Annabella Drewanowski[3], Pascal Rothenbühler[3], Jaewon Choi [8], Ronny Sutarto [9], Yao Wang [10], Tao Xiang [1,6,11], Jiangping Hu [1,6], Ke-Jin Zhou [8], Valentina Bisogni [2], Riccardo Comin[5], J. Chang [3] ✉, Jonathan Pelliciari [2] ✉, X. J. Zhou [1,6,12] ✉ & Zhihai Zhu [1,6,12] ✉

Strongly correlated materials respond sensitively to external perturbations such as strain, pressure, and doping. In the recently discovered superconducting infinite-layer nickelates, the superconducting transition temperature can be enhanced via only ~1% compressive strain-tuning with the root of such enhancement still being elusive. Using resonant inelastic x-ray scattering (RIXS), we investigate the magnetic excitations in infinite-layer PrNiO$_2$ thin films grown on two different substrates, namely SrTiO$_3$ (STO) and (LaAlO$_3$)$_{0.3}$(Sr$_2$TaAlO$_6$)$_{0.7}$ (LSAT) enforcing different strain on the nickelates films. The magnon bandwidth of PrNiO$_2$ shows only marginal response to strain-tuning, in sharp contrast to the enhancement of the superconducting transition temperature $T_c$ in the doped superconducting samples. These results suggest the bandwidth of spin excitations of the parent compounds is similar under strain while $T_c$ in the doped ones is not, and thus provide important empirics for the understanding of superconductivity in infinite-layer nickelates.

High-temperature superconductivity continues to be a challenging topic in correlated quantum matter since multiple electronic phases emerge in proximity to each other, masking the leading interaction for electron pairing. The newly discovered superconducting infinite-layer nickelates provide a new platform to study unconventional superconductivity[1]. A central question that has soon arisen for these systems is to what extent they are analogs of cuprate superconductors. Understanding the similarities and distinctions between these two families of materials may help bring to light new aspects of high-$T_c$

superconductivity, and in particular, the pairing mechanism[2]. Recent experiments have revealed significant differences between these two classes of materials. For instance, cuprates are charge-transfer insulators, while the parent compounds of infinite-layer nickelates are likely Mott-Hubbard type[3], or somewhere between the charge-transfer and Mott-Hubbard regime[4] according to the Zaanen–Sawatzky–Allen (ZSA) scheme[3–7]; unlike the cuprates, the rare-earth spacing layers in infinite-layer nickelates hybridize with Ni 3$d$ orbitals, leading to 5$d$ metallic states at the Fermi level[3,5]. Despite these differences, infinite-

[1]Beijing National Laboratory for Condensed Matter Physics, Institute of Physics, Chinese Academy of Sciences, Beijing 100190, China. [2]National Synchrotron Light Source II, Brookhaven National Laboratory, Upton, New York, NY 11973, USA. [3]Physik-Institut, Universität Zürich, Winterthurerstrasse 190, CH-8057 Zürich, Switzerland. [4]Department of Physics, The Chinese University of Hong Kong, Shatin, Hong Kong, China. [5]Department of Physics, Massachusetts Institute of Technology, Cambridge, MA 02139, USA. [6]University of Chinese Academy of Sciences, Beijing 100049, China. [7]AGH University of Science and Technology, Faculty of Physics and Applied Computer Science, 30-059 Kraków, Poland. [8]Diamond Light Source, Harwell Campus, Didcot OX11 0DE, United Kingdom. [9]Canadian Light Source, Saskatoon, Saskatchewan S7N 2V3, Canada. [10]Department of Physics and Astronomy, Clemson University, Clemson, SC 29631, USA. [11]Beijing Academy of Quantum Information Sciences, Beijing 100193, China. [12]Songshan Lake Materials Laboratory, Dongguan 523808, China. [13]These authors contributed equally: Qiang Gao, Shiyu Fan, Qisi Wang. ✉e-mail: johan.chang@physik.uzh.ch; pelliciari@bnl.gov; XJZhou@iphy.ac.cn; zzh@iphy.ac.cn

layer nickelates share several general characteristics with cuprates, including a linear temperature dependence of resistivity for the normal state[8], a dome-like shape for $T_c$ as a function of doping in the phase diagram[9–14], a sizable magnetic exchange interaction[15–18], possible charge density wave instabilities[17–19], and a possible $d$-wave superconducting gap[20–22]. These properties corroborate that the superconductivity in infinite-layer nickelates is unconventional.

An approach to tackling the pairing mechanism is to directly manipulate $T_c$ with controllable knobs, and simultaneously examine the response of the bosonic excitations to these perturbations. By examining the coupling between low-energy excitations and external tuning parameters, one may identify the leading interaction channels accounting for the superconductivity. The use of different substrates has been proposed as a pathway for tuning $T_c$ in thin films of infinite-layer nickelates[1]. However, it is highly challenging to obtain superconducting films on substrates other than $SrTiO_3$ (STO). Nevertheless, recent experiments have shown an increase of $T_c$ by 40% for the $Pr_{0.8}Sr_{0.2}NiO_2$ films grown on $(LaAlO_3)_{0.3}(Sr_2TaAlO_6)_{0.7}$ (LSAT) (with respect to STO)[23] therefore calling for a connection with strain. Similarly, enhancing $T_c$ through strain engineering has been observed in $La_{2-x}Sr_xCuO_4$ thin films grown on different substrates[24–26], where the increase of $T_c$ is attributed to strengthening the magnetic exchange interaction by compressive strain[27]. Although long-range magnetic ordering has not been found in infinite-layer nickelates to date, resonant inelastic x-ray scattering (RIXS) studies on $Nd_{1-x}Sr_xNiO_2$ have revealed propagating spin excitations, resembling the ones of the spin-1/2 antiferromagnet (AFM) on the square lattice, with a large spin exchange energy ~60 meV in the parent compound[15]. Presently, how the magnetic excitations couple to strain in nickelates is still unexamined due to the limited scattering volume precluding inelastic neutron scattering.

Here, using high-resolution Ni $L_3$-edge RIXS, we explored the role of the substrate on the magnetic excitations in infinite-layer $PrNiO_2$ thin films. We observe spin excitations regardless of the strain level.

Moreover, the substrate tuning shows a marginal influence on the bandwidth of magnon dispersion, in contrast to the enhancement of $T_c$ for superconducting films. These results suggest that the energy scale of spin fluctuations in parent compounds exerted with different strain values does not directly correlate with $T_c$, at odds with what has been reported in cuprates[27–29]. Considering that the magnetism of the parent compound is the starting point to reach superconductivity upon doping, our results provide important implications for elucidating the electron pairing mechanism in superconducting infinite-layer nickelates.

## Results

Figure 1a shows a schematic of the scattering geometry of our RIXS measurements. It has been shown that the magnetic excitations in cuprates can be detected with RIXS by using either grazing-in geometry with σ (linear vertical) incident light polarization or grazing-out geometry with π (linear horizontal) polarized incident photons[30]. We adopted the former for the majority of the data in the present study, which allowed for the detection of both magnetic excitations and phonons. The x-ray absorption spectra (XAS) of the $PrNiO_2$ films display a much stronger absorption peak in σ than π polarization at the Ni $L_3$ edge ($2p^6 3d^9$–$2p^5 3d^{10}$ transition) (see Fig. 1c, d). This linear dichroism reflects the $d_{x^2-y^2}$ symmetry of the 3d hole in $PrNiO_2$. In the $PrNiO_2$ film on LSAT, the strong signal at ~850 eV is associated with the La $M_4$ edge ($3d - 4f$ transition) in the LSAT substrate. A clear linear dichroism is observed at both the La $M_4$ and Ni $L_3$ absorption edges. A comparable linear dichroism at the Ni $L_3$ absorption edge corroborates that the films on both STO and LSAT have equivalent sample quality.

Moving to the RIXS spectra, the high energy ($dd$) excitations provide valuable information regarding the local configurations of the 3d orbitals in Ni ions which is determined by the symmetry of the crystal field. In infinite-layer nickelates, the expected $D_{4h}$ crystal field leads to the splitting of the Ni 3d orbitals with the $d_{x^2-y^2}$ at the highest energy, followed in sequence by the $d_{xy}$, $d_{yz/xz}$, and $d_{3z^2-r^2}$, as illustrated

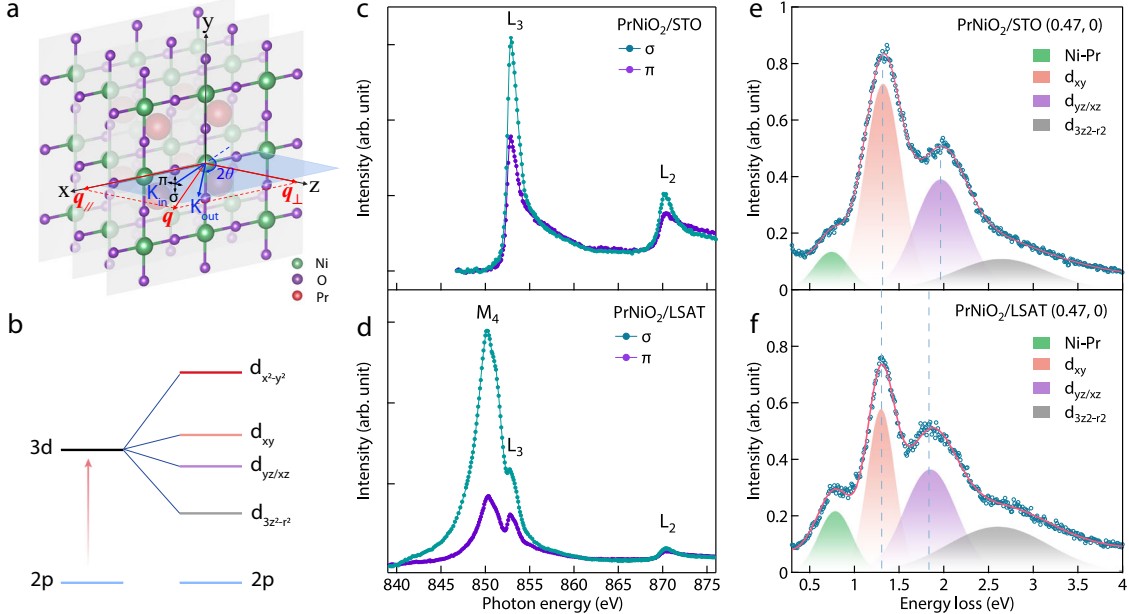

**Fig. 1 | X-ray absorption spectra (XAS) and high energy ($dd$) excitations of $PrNiO_2$ films grown on STO and LSAT. a** Crystal structure of $PrNiO_2$ and scattering geometry of the RIXS experiments. The polarization of the incoming photon is fixed to σ or π, where σ and π represent, respectively, the polarization components, perpendicular and parallel to the scattering plane. The 2θ scattering angle is fixed at 150° (or 154°) to maximize the in-plane momentum transfer, which is tuned by rocking the sample. $q_{//}$ ($q_\perp$) refers to the momentum transfer that is parallel (perpendicular) to the nickel-oxide plane, respectively. **b** The $d$-level splitting of Ni ion in the $D_{4h}$ crystal field. **c, d** The XAS of the $PrNiO_2$ films grown on STO and LSAT substrate measured by σ and π polarization. All the XAS measurements were performed with the grazing-in geometry at an incident angle of 30 degrees. **e, f** The high energy $dd$ excitations of the $PrNiO_2$ films grown on STO and LSAT at a representative momentum, respectively. The blue dashed lines represent the peak positions of the $d_{xy}$ and $d_{yz/xz}$ orbital excitations.

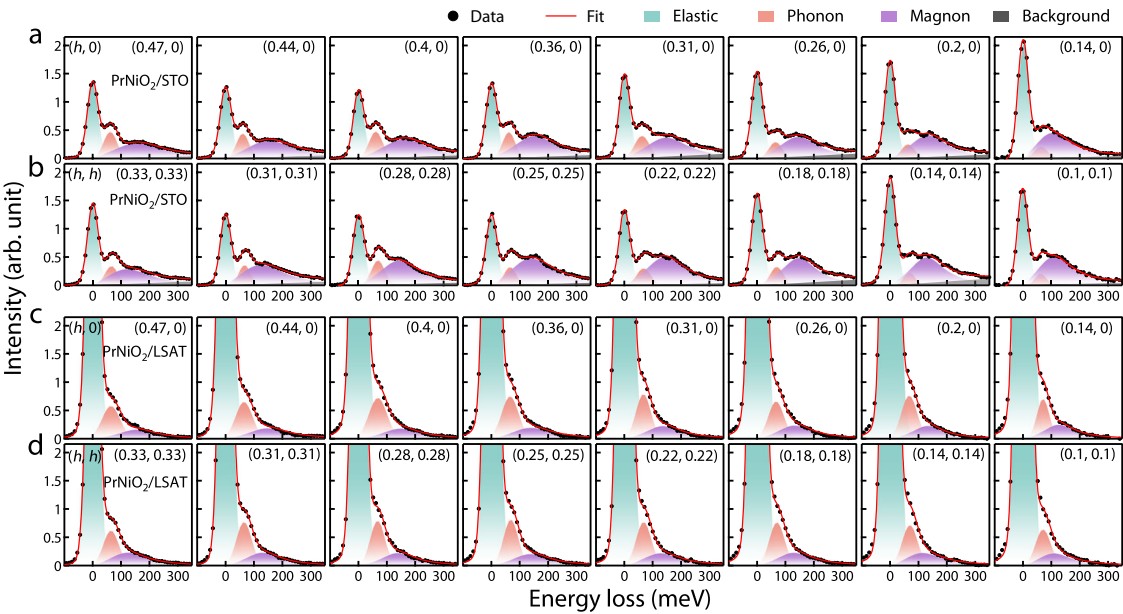

**Fig. 2 | Momentum resolved RIXS spectra along high symmetry directions.**
**a**, **b** RIXS spectra of the PrNiO$_2$ film grown on STO along $(h, 0)$ and $(h, h)$ directions. **c**, **d** RIXS spectra of the PrNiO$_2$ film grown on LSAT along $(h, 0)$ and $(h, h)$ directions. The filled black circles represent the data and the solid red curves fit the data set, using a combination of an elastic scattering contribution (green), a Gaussian profile for the phonon peak (orange), a DHO function for the magnetic excitation (purple), and background (gray). All the measurements were taken at 40 K.

in Fig. 1b. We show in Fig. 1e, f the $dd$ excitations in the RIXS spectra of PrNiO$_2$ on STO and LSAT, respectively. Both spectra exhibit four major features marked by the shaded areas. The spectral peaks in the energy-loss range of 1–4 eV correspond to the crystal field splitting as illustrated in Fig. 1b, and the peak at ~0.7 eV arises from the hybridization between Ni and Pr ions, which is similar to studies on ANiO$_2$ (A= La, Nd)[3,15,17–19,31]. As denoted by the blue dashed lines, the peaks assigned to the transition to the $d_{xy}$ orbital are comparable for both samples regarding the peak center of mass positions as well as line shapes; the peak assigned to the transition to $d_{yz/xz}$ orbital moves toward lower energy for PrNiO$_2$/LSAT compared with PrNiO$_2$/STO. The studies on various cuprate families have revealed a power law relationship $E_{xy} \propto a^{-4}$ between the $E_{xy}$ state energy referring to the $x^2 - y^2$ ground state and the in-plane lattice constant $a$[32]. This simple relation might also be applied to the case of infinite-layer nickelates and would yield an increase of 4% in $E_{xy}$ ~0.05 eV, which is hard to resolve in the broad RIXS spectra for $dd$ excitations. This explains the nearly equal $E_{xy}$ state energies for the films on both substrates, as shown in Fig. 1e, f. The transition to the $d_{yz}/d_{xz}$ orbital shifts toward lower energy for PrNiO$_2$/LSAT (1% compressive strain) in comparison to PrNiO$_2$/STO, in contrast to the observations made on cuprate films[27]. A possible picture to reconcile the inconsistency is that Ni-O-Ni bond angles deviate from 180°, and this deviation is further enhanced under compressive strain, leading to the decrease of the absolute value of the $x^2 - y^2$ orbital energy. Qualitatively, the energy of xz/yz orbital ($pd\pi$ overlap) is less affected than that of $x^2 - y^2$ orbital ($pd\sigma$ overlap) by compressive strain, thus the energy shift in $E_{xz/yz}$ is dominated by the $x^2 - y^2$ orbital energy. All in all, the discernible changes of $dd$ excitations suggest that the electronic structures of PrNiO$_2$ are effectively modified by the epitaxial strain variation which is confirmed by the reciprocal space maps (RSM) collected around the (103) reflections for the PrNiO$_2$ films on the different substrates LSAT and STO (see the supplementary materials for details).

In Fig. 2, we show the low-energy loss RIXS spectra for PrNiO$_2$/STO and PrNiO$_2$/LSAT along high symmetry directions $(h, 0)$ and $(h, h)$ in momentum space. The spectra of PrNiO$_2$/STO consist of three major features composed of an elastic peak at zero energy loss, a clear

excitation at ~60 meV, and a broad peak in the range of 100–400 meV. The peak at ~60 meV represents the phonon excitation, reminiscent of the ~70 meV phonon mode that prevails in cuprates[15,33]. The broad peak is ascribed to the magnetic excitations as it disperses as a function of momentum, resembling what was observed in NdNiO$_2$ and LaNiO$_2$[15,17,18]. In PrNiO$_2$/LSAT, the elastic peak is more prominent, probably owing to the contribution from the La $M_4$ edge in the substrate. Nevertheless, the excitations can still be reliably extracted by fitting the RIXS spectra to a combination of a Voigt function for the elastic peak, a Gaussian function for the phonon, a damped harmonic oscillator (DHO) to account for the magnetic excitation, and a smoothly varying background. The DHO function $\chi''(q, \omega)$ is given by

$$\chi''(q,\omega) = \frac{\gamma_q \omega}{\left(\omega^2 - \varepsilon_q^2\right)^2 + 4\gamma_q^2 \omega^2} \tag{1}$$

where $\varepsilon_q$ is the undamped mode energy, and $\gamma_q$ is the damping factor[30]. As shown in Fig. 2, the fitting overall describes well the experimental spectra.

To better represent the magnon dispersion, we include in Fig. 3a, b the magnetic spectra map of the magnetic excitations after subtracting the elastic peak, phonon peak, and background from the raw data. As shown in Fig. 3a, a clear magnon dispersion can be visualized along both $(h, 0)$ and $(h, h)$ directions for PrNiO$_2$/STO, while in the case of PrNiO$_2$/LSAT (see Fig. 3b), the magnetic excitations appear to be weak and less dispersive. To directly characterize the response of magnetic excitations to strain-tuning, we show in Fig. 3c–f the RIXS spectra at the zone boundaries. Here π incident light polarization and grazing-out scattering geometry were used to enhance the magnon intensity. Again, the DHO function was used to fit the magnon peaks. The gray dashed lines represent the magnon energies from the fits, while the blue dashed lines denote the energies of peak maximum. The magnons move toward lower energies in PrNiO$_2$/LSAT compared with PrNiO$_2$/STO.

Figure 4a shows the momentum dependence of the fitted values of the magnetic excitation energy $\varepsilon_q$ and the damping factor $\gamma_q$ as

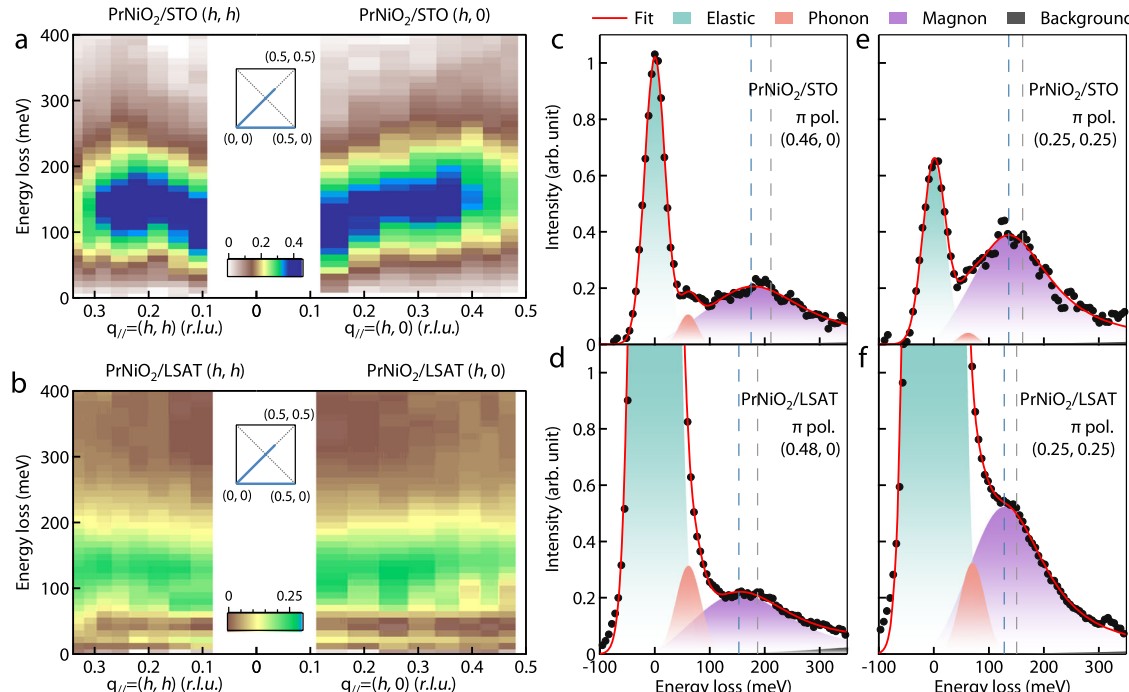

**Fig. 3 | Dispersion of the magnetic excitations in PrNiO₂. a, b** RIXS intensity map of the PrNiO₂ films grown on SrTiO₃ (**a**) and LSAT (**b**) along (h, h) and (h, 0) directions at 40 K, which were obtained by subtracting the elastic peak, phonon, and the background components for better visualization. The insets in (**a, b**) show the trajectory in momentum space of the RIXS measurements. **c–f** RIXS spectra of the PrNiO₂ film grown on STO and LSAT at the zone boundaries measured by π polarization. The filled black circles represent the data and the solid red curves fit the data set, using a combination of an elastic scattering contribution (green), a Gaussian profile for the phonon peak (orange), a DHO function for the magnetic excitation (purple), and a smoothly varying background (gray). The gray dashed lines represent the mode energy ($\varepsilon_q$) of the magnetic excitations obtained by fitting with the DHO function and the blue dashed lines represent the peak energy of the magnon.

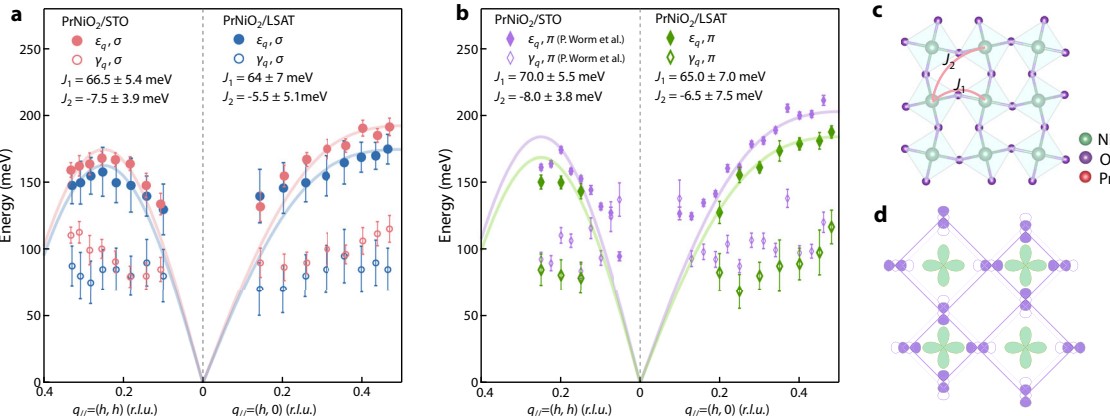

**Fig. 4 | Dispersion of the magnetic excitations in PrNiO₂ and comparison to the model calculations. a, b** The mode energy ($\varepsilon_q$) and damping factor ($\gamma_q$) in PrNiO₂ grown on STO and LSAT obtained from fitting the RIXS spectra of PrNiO₂ grown on STO (LSAT) measured by σ polarization (**a**) and π polarization (**b**); the data set for the PrNiO₂ grown on STO in (**b**) is from ref. 35. The solid lines represent the best-fit for the model of the spin-1/2 Heisenberg antiferromagnet on the square lattice using linear-spin-wave theory. $J_1$ and $J_2$ represent the nearest-neighbor (NN) and next-nearest-neighbor (NNN), respectively. Error bars are estimated from the standard deviation obtained by the least-squares fitting algorithm and multiple times of fittings. **c** Illustration of possible structural distortions in the PrNiO₂ films. **d** Schematic plot of the bond-stretching phonon modes that may suppress the superexchange coupling strength.

defined in Eq. (1) for both samples. Considering the uncertainty for the fitting, the magnon dispersions are comparable in both cases, with an energy maximum close to (0.5, 0) along (h, 0) direction, and close to (0.25, 0.25) along (h, h) direction; they are similar to the magnon dispersions for the spin-1/2 Heisenberg AFM on the square lattice. The magnon bandwidth in PrNiO₂/LSAT appears to be slightly reduced compared to that in PrNiO₂/STO. The damping factors of the two samples are comparable and vary little in both directions. To model the energy scale of the spin excitations, we fit the extracted magnetic dispersion by resorting to linear-spin-wave theory[34]. The Hamiltonian is given by

$$H = J_1 \sum_{\langle ij \rangle} S_i \cdot S_j + J_2 \sum_{\langle ii' \rangle} S_i \cdot S_{i'} \qquad (2)$$

where $S_i$ is the spin-1/2 operator on site $i$, and $\langle ij \rangle$ ($\langle ii' \rangle$) denotes the nearest neighbors (next-nearest neighbors). The best-fit to the spectra

yields $J_1$ ($J_2$) = 66.5 (−7.5) meV for PrNiO$_2$/STO, and $J_1$ ($J_2$) = 64 (−5.5) meV for PrNiO$_2$/LSAT, similar to the findings for NdNiO$_2$ on STO[15]. The magnon dispersions near the zone center exhibit a noticeable deviation from the fits to the linear-spin-wave dispersion (the solid lines). This is caused by the substantial overlapping spectra peaks of the magnon, phonon, and elastic signals near the zone center, leading to a relatively larger error in determining the magnon peak positions. All in all, the spin exchange coupling for PrNiO$_2$/LSAT is nearly equal to that for PrNiO$_2$/STO, suggesting that the in-plane compressive strain of -1% has a marginal influence on the superexchange coupling $J$. Figure 4b presents an additional data set obtained using π (linear horizontal) polarized incident photons, where the one for PrNiO$_2$/STO is from the reference and obtained on our sample[35]. A similar analysis using linear-spin-wave theory yields $J_1$ ($J_2$) = 70.0(−8.0) meV for PrNiO$_2$/ STO, and $J_1$ ($J_2$) = 65 (−6.5) meV for PrNiO$_2$/LSAT, suggesting the magnon bandwidth is slightly reduced in response to the in-plane compressive strain of -1% enforced by LSAT referring to STO. This is consistent with the result of the measurements using σ polarized incident photons.

## Discussion

In the RIXS studies on La$_2$CuO$_4$ thin films, the Coulomb and magnetic exchange interactions are strengthened by the compressive strain imposed by the substrates, which may account for the doubling of $T_c$ in the doped films[27]. Empirically, the superexchange $J$ in an insulator is expected to scale with inter-ion distance $a$ by $J \sim a^{-10}$ [36]. This would approximately lead to a -10% enhancement of $J$ for 1% compressive strain. First-principles calculations on this system predict that the magnon bandwidth increases by 7.8% for a −1% strain[23], corresponding to -15 meV. From our results, the energy scale of the spin excitations, which is determined by the superexchange interaction between the planar nearest-neighbor Ni spins, seems to be slightly reduced when a -1% compressive strain is applied. Admittedly, the small increase in the magnon bandwidth could be obscured considering the uncertainty in the fitting of the magnon dispersions to a linear-spin-wave model. However, as shown in Fig. 4a, b, the spin excitations in PrNiO$_2$/LSAT are systematically lower rather than higher in energy compared with those in PrNiO$_2$/SrTiO$_3$, demonstrating a small but nonetheless decrease of superexchange $J$ under compressive strain. Besides, as shown in Fig. 3c−f, the RIXS spectra with well-defined magnon peaks at the zone boundaries show that the energies of spin excitations are of the same order of magnitude and, likewise, a small decrease in PrNiO$_2$/LSAT compared with PrNiO$_2$/SrTiO$_3$.

A possible explanation for this difference in comparison with La$_2$CuO$_4$ is that there may exist a structural distortion of the Ni-O plane, which would modify the bond angle of Ni-O-Ni under compressive strain, and hence the superexchange $J$. As illustrated in Fig. 4b, such structural distortion is commonly seen in the precursor phase $R$NiO$_3$ ($R$ is a rare-earth element such as La, Pr, Nd, and Sm)[37,38]. After the topotactic transformation, the structural distortion may exist in the infinite-layer phase[39]. Similarly, structural distortion with modified bond angles has been proposed to explain the stripe-like charge ordering in La$_4$Ni$_3$O$_8$ and La$_3$Ni$_2$O$_6$[40]. Alternatively, the lattice fluctuation induced by the electron-phonon interaction suppresses the effective superexchange $J$, as recently proposed for parent compounds of cuprates[41]. In this scenario, bond-stretching phonons dynamically drag oxygen away from its equilibrium position, as illustrated in Fig. 4c; accordingly, the superexchange strength would be suppressed as the electron-phonon coupling increases under compressive strain. However, such a picture has been demonstrated only for one-dimensional (1D) systems, and whether it can be generalized to 2D systems remains unclear. In any case, this reduced magnetic exchange coupling due to 1% compressive strain in infinite-layer nickelates differs from that in the La$_2$CuO$_4$ thin films mentioned above[27].

For unconventional superconductivity, spin fluctuations are the main candidate as the driving force for condensing electrons into pairs[42]. In this scenario, spin interactions primarily set the energy scale for superconductivity, which occurs in close proximity to an anti-ferromagnetic phase. In the large $U$ limit, $T_c$ is expected to scale with the superexchange $J$ at the mean-field level for studies on cuprate superconductors[28,43]. This has also been used to estimate $T_c$ for the potential LaNiO$_3$/LaMO$_3$ superconductors, where M = Al, Ga, and Ti[44]. In this context, an enhancement of -27 meV in magnon bandwidth would be expected to scale with an increase of $T_c$ by ~40%, which, however, contradicts our experimental observations, at odds with what has been recently reported for cuprates[27–29]. Admittedly, it is possible that besides the electronic modification via strain, other effects such as crystalline quality, interface reconstruction, and partial relaxation in the film may be at play as well[8,45]. This warrants further study to disentangle individual contributions to the enhancement of $T_c$ as well as to the magnetic excitations[45]. We notice that freestanding superconducting infinite-layer nickelate membranes have recently been reported[46,47], which offer opportunities for achieving even larger strain variation without the complexity from the sample quality or interface between film and substrates. Our study will motivate further exploration of superconductivity in infinite-layer nickelates by monitoring low-energy excitations versus strain variation using more different substrates[33,48–58].

## Methods
### Sample preparation

Thin films of the precursor phase PrNiO$_3$ with a thickness of -7 nm were prepared by using pulsed laser deposition (PLD) on (001)-oriented SrTiO$_3$ and LSAT substrates with a 248-nm KrF excimer laser[23,59]. The infinite-layer phase PrNiO$_2$ was obtained by a soft-chemistry reduction process using CaH$_2$ powder. Substrates were pre-annealed at 900 °C with an oxygen partial pressure of $1 \times 10^{-5}$ Torr. During growth, the substrate was kept at 600 °C under an oxygen partial pressure of 150 mTorr. After deposition, the films were cooled to room temperature at a rate of 5 °C per minute in the same oxygen partial pressure. For CaH$_2$ topotactic reduction, the as-grown nickelate films were sealed with 0.1 g CaH$_2$ powder and annealed at a temperature of -290 °C for -3 h. After reduction, the PrNiO$_2$ films were loaded back to the PLD chamber and capped with an amorphous SrTiO$_3$ layer of 10 nm at room temperature to protect the films. After the reduced films were capped by SrTiO$_3$, XRD measurements were carried out to confirm the high quality of the infinite-layer phase (see Supplementary Fig. S1).

### XAS and RIXS measurements

The X-ray absorption spectroscopy (XAS) measurements at the Ni $L$-edge were performed at the resonant elastic inelastic X-ray scattering (10-ID2) of the Canadian Light Source, equipped with a 4-circle diffractometer in a $10^{-10}$ mbar ultrahigh-vacuum chamber. The photon flux is about $5 \times 10^{11}$ photons per second and energy resolution reaches $\Delta E/E$ - $2 \times 10^{-4}$. The incoming photons are fully polarized with two configurations linear vertical (σ) and horizontal (π).

High-resolution resonant inelastic X-ray scattering (RIXS) measurements were mainly performed at the SIX 2-ID beamline of NSLS-II using σ incident light polarization. The energy resolution was set to $\Delta E = 34$ meV (full-width-at-half-maximum) at the Ni $L_3$ edge[60]. Additional RIXS spectra (Figs. 3c−f, 4b) were taken with π incident light polarization at the I21 beamline at the Diamond Light Source, where the energy resolution was set to 39 and 54 meV for measurements on PrNiO$_2$/STO and PrNiO$_2$/LSAT, respectively[61]. The SIX (I21) spectrometer was positioned at the largest scattering angle of 150° (154°) to maximize the in-plane momentum transfer, and the sample temperature was set to 40 (16) K. All RIXS spectra are normalized to the area of the $dd$ excitations (400–4000 meV).

## Data availability

The article and its Supplementary Information files contain all the data needed to evaluate the conclusions in the paper. All data generated in this study have been deposited in the Figshare database, which is open access at https://doi.org/10.6084/m9.figshare.26003392.v1.

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

## Acknowledgements

We thank Fu-Chun Zhang, Shiliang Li, Yi Zhou, and Yuan Wan for fruitful discussions. This work was supported in part by the National Natural Science Foundation of China (Grant No. 12074411) and (Grant No. 11888101), the National Key Research and Development Program of China (Grant No. 2016YFA0300300 and 2017YFA0302900), the Synergetic Extreme Condition User Facility (SECUF), the Strategic Priority Research Program (B) of the Chinese Academy of Sciences (Grant No. XDB25000000) and the Research Program of Beijing Academy of Quantum Information Sciences (Grant No. Y18G06). J.L. and R.C. acknowledge support from the Air Force Office of Scientific Research Young Investigator Program under grant FA9550-19-1-0063. Q.W. is supported by the Research Grants Council of Hong Kong (ECS No. 24306223). Q.W. and J.C. acknowledge support by the Swiss National Science Foundation. Work at Brookhaven National Laboratory was supported by the US Department of Energy (DOE) Office of Science under Contract No. DE-SC0012704, Early Career Research Program, and the Laboratory Directed Research and Development project of Brookhaven National Laboratory under Contract No. 21-037. We acknowledge the I21-RIXS Beamline for providing beamtime under Proposal MM30189. Part of the research described in this paper was performed at the Canadian Light Source, a national research facility of the University of Saskatchewan, which is supported by the Canada Foundation for Innovation (CFI), the Natural Sciences and Engineering Research Council (NSERC), the National Research Council (NRC), the Canadian Institutes of Health Research (CIHR), the Government of Saskatchewan, and the University of Saskatchewan.

## Author contributions

J. Chang., X.J.Z., and Z.H.Z. conceived the research. Q.G. and X.L.R. prepared and characterized the film samples. X.L.R. and R.S. performed XAS measurements. J.P., S.Y.F., Q.G., J.R.L., Z.H.Z., R.C., Q.S.W., I.B., A.D., P.R., J. Choi, and J. Chang performed the RIXS experiments with the help of V.B. and K.J.Z. Q.G., Z.H.Z., and Q.S.W. analyzed the data. Y.W., T.X., and J.P.H. provided theoretical understanding. Q.G. and Z.H.Z. wrote the manuscript with input from all authors.

## Competing interests

The authors declare no competing interests.
