## [Peer Review File · Nature Communications]

Reviewers' Comments:

Reviewer #1:

Remarks to the Author:

I read the reply of the authors and looked at the changes made to the manuscript.

Most of the answers and proposed changes are fine except for my most important comment on the role of strain.

The authors cite Lee et al, Nature 619 (2023) but they have not read the supplementary information carefully – see <https://www.nature.com/articles/s41586-023-06129-x#Sec15>. They should look at Figure S4b.

I understand the authors are saying that the change from the gray to the red dome is consistent with what they see because it represents an enhancement in T_c when changing from STO to LSAT. The important point is that the purple dome is the new growth conditions on STO, in other words the same conditions as on LSAT. Around optimal doping there is no perceptible change in T_c .

It is good that they have added mention of the possibility that the substrate introduces other effects, like crystallinity, that may be changing the T_c . But I think the authors have to acknowledge that other groups are not seeing the same T_c enhancement suggesting that we are still in a regime where there is considerable sample-to-sample scattering of T_c making it difficult to resolve a clear strain effect.

My second comment regards the RSM on LSAT - new Figure S1f. It looks like the PSNO is partially relaxed, as quite a bit of peak weight can be seen "on the left" of the (103) LSAT peak. This portion has roughly the same in-plane lattice constant as the strained film on STO. It is hard to tell what fraction of the film contributes to this intensity. Can the authors quantify it? The relaxation is about 0.7%, so not fully relaxed to the nickelate bulk, but this might be important since the nominal strain difference between LSAT and STO is only 1%.

In summary, it is certainly plausible that there could be a strain dependence of T_c but the present study cannot make such a strong claim with only two samples, one of which is partially relaxed, and contrasting results from other groups. The data seems high quality and, as I said before, should definitely be published. The conclusions drawn are simply too strong to be supported.

Reviewer #2:

Remarks to the Author:

This is my second review of the manuscript entitled "Magnetic Excitations in Strained Infinite-layer Nickelate PrNiO₂ Films" which has now been transferred to Nature Communications.

I thank the authors for their detailed response and I recognize that the authors have put some work to improve the manuscript, especially by performing new experiments and adding supporting data.

The authors have addressed some of my previous comments in their response and provided corresponding modifications in the manuscript. However, I still have concerns that are related to points (4) and (5) of my previous review, that prevent my recommendation for publication of this manuscript in Nature Communications in its present form.

I detail below my concerns:

- To me, the raw data that clearly support the authors' conclusions are the measurements presented on Fig. 3 c-f, that are the RIXS spectra recorded with π -polarization on both PrNiO₂/STO vs PrNiO₂/LSAT. In these data, the magnetic excitation is clearly separated from the elastic peak for both samples and the data analysis seem robust. However, these concern only two

Q points (0.48, 0) and (0.25, 0.25) and therefore this is not sufficient to determine a dispersion. The data measured with sigma-polarization on PrNiO₂/LSAT (Fig. 2 c-d) that serve to determine the dispersion plotted on Fig. 4 a, on the other hand, show a weak magnetic excitation signal that is more a shoulder of the elastic peak and phonon contributions than a clear peak. I am still troubled by the weakness of the magnetic peak signal measured with sigma-polarization on PrNiO₂/LSAT (Fig. 2 c-d) compared to PrNiO₂-STO (Fig. 2 a-b). The authors did not give a clear explanation as to why the magnetic peak is weak in these measurements. This is not the case for the measurements using pi-polarization, where both measurements on the two different samples show similar intensity for the magnon (Fig. 3 c-f). The authors should explain why the magnetic peaks in PrNiO₂-LSAT are weak in comparison to the PrNiO₂-STO sample on the sigma-polarization data (Fig. 2).

- I am also still troubled by the extracted position of the magnetic excitations on both PrNiO₂-LSAT and PrNiO₂-STO samples that changes substantially between measurements with sigma-polarization and those with pi-polarization, i.e. red circles and blue squares vs red and green diamonds on Fig. 4 a (almost 25 meV difference for PrNiO₂-STO). While I understand that at the end the results are equivalent between the two pi and sigma measurements, I however wonder if the authors have an explanation for such difference between pi and sigma measurements? As the authors stated in their response, the spectra can be different between sigma and pi ("larger contrast between elastic / phonon and magnetic excitations, and weaker spectra distorted by self-absorption"), but are not supposed to give different positions for the extracted peaks. Does this difference affect the error determination of the magnetic peak position and estimation of the super-exchange coupling J?

- Considering the data quality shown on Fig. 2 c-d, I suggest to plot these data and fit contributions with a zoom on the interesting part, i.e. plotting with a smaller intensity scale for instance or by changing the aspect ratio in order to better highlight the spectra and corresponding fits near the magnetic peak, or at least to show such zoom in the Supplementary Information. With the current plotting format, the reader cannot judge the quality of the fit, especially on the tail of the magnetic peak for the PrNiO₂/LSAT sample.

- Again, considering the data quality shown on Fig. 2 c-d, I still have concerns about the fitting model of the PrNiO₂/LSAT RIXS spectra and therefore wonder how accurate are the extracted magnetic peak positions along with the corresponding error bars plotted on Figure 4 (a). What are the results of the fits on the PrNiO₂/LSAT sample (Fig. 2 c-d) if the magnon peak position is fixed to the expected value to scale with an increase of T_c by 40%? The authors should perform this counter analysis and compare to the current model.

Reply to Reviewer #2

We thank the Reviewer for the thorough assessment of our manuscript. We are grateful for the evaluation and are glad that the Reviewer finds “*The data seems high quality and, as I said before, should definitely be published.*”

Below we address the questions raised by the Reviewer. We indicate in **blue** the original reports, in **black** our reply, and in **red** our changes in the revised manuscript.

(1) Reviewer’s question: “I read the reply of the authors and looked at the changes made to the manuscript. Most of the answers and proposed changes are fine except for my most important comment on the role of strain. The authors cite Lee et al, *Nature* 619 (2023) but they have not read the supplementary information carefully see **Figure S4b**. They should look at **Figure S4b** I understand the authors are saying that the change from the gray to the red dome is consistent with what they see because it represents an enhancement in T_c when changing from STO to LSAT. The important point is that the purple dome is the new growth conditions on STO, in other words the same conditions as on LSAT. Around optimal doping there is no perceptible change in T_c .

It is good that they have added mention of the possibility that the substrate introduces other effects, like crystallinity, that may be changing the T_c . But I think the authors have to acknowledge that other groups are not seeing the same T_c enhancement suggesting that we are still in a regime where there is considerable sample-to-sample scattering of T_c making it difficult to resolve a clear strain effect.”

Our response: We thank the Reviewer for kindly reminding us of the right figure to read. We agree with the Reviewer that **Figure S4b** indeed does not show a perceptible change in T_c around optimal doping in $\text{Nd}_{1-x}\text{Sr}_x\text{NiO}_2$. We were aware that more recently freestanding superconducting membranes of $(\text{Nd},\text{Sr})\text{NiO}_2$ have been reported in arXiv:2402.05104 by Hwang’s group, showing the signature of enhancement of T_c by compressive strain. This is also consistent with another report in $(\text{La},\text{Sr})\text{NiO}_2$ (arXiv:2402.05104). Nevertheless, we acknowledge that at the current stage of this field, we still have variations in sample details that prevent us from resolving a clear strain effect. As also praised by the reviewer, we have toned down our conclusion by considering other effects such as crystalline quality and interface details, which may account for the variation in T_c .

(2) Reviewer’s question: “My second comment regards the RSM on LSAT - new **Figure S1f**. It looks like the PSNO is partially relaxed, as quite a bit of peak weight can be seen “on the left” of the (103) LSAT peak. This portion has roughly the same in-plane lattice constant as the strained film on STO. It is hard to tell what fraction of the film contributes to this intensity. Can the authors quantify it? The relaxation is about 0.7%, so not fully relaxed to the nickelate bulk, but this might be important since the nominal strain difference between LSAT and STO is only 1%.

In summary, it is certainly plausible that there could be a strain dependence of T_c but the present study cannot make such a strong claim with only two samples, one of which is partially relaxed, and contrasting results from other groups. The data seems high quality and, as I said before, should definitely be published. The conclusions drawn are simply too strong to be supported.

Our response: We thank the reviewer for raising the concern on the details of the RSM

of the (103) peak. We agree with the reviewer that some peak weight exists on the left of the (103) LSAT peak, suggesting the film is partly relaxed. It is indeed very hard to quantify the volume fraction of the broadening of the reflection by using X-ray diffraction on a thin film since the intensity of X-ray diffraction seems not directly linked to the volume fraction. Maybe high-resolution STEM can be employed to reveal the volume fraction or the gradient of the relaxation, which is however beyond our expertise. We would like to leave this question to our future study. We thank the reviewer for pointing this out and acknowledge that it should be considered in drawing a proper conclusion. Therefore, we have added in the supplementary Note 1: “**It is noticeable that some broad peak weight exists in the reciprocal space maps (RSM) of the (103) reflection in Fig. S1f, indicating the film is partly relaxed.**” Besides, we have softened our conclusion by acknowledging the contrasting results in the literature as well as the partial relaxation of the film as pointed out by the reviewer. We have also emphasized in the second paragraph on page 9 by stating that “**Admittedly, it is possible that besides the electronic modification via strain, other effects such as crystalline quality, interface reconstruction, and partial relaxation in the film may be at play as well [8, 45].**”

Changes made: We have used a rather measured tone on the enhancement of T_c versus strain considering contrasting results across the groups, and have further softened our conclusion in the revised manuscript by emphasizing in the third paragraph on page 9: “**Admittedly, it is possible that besides the electronic modification via strain, other effects such as crystalline quality, interface reconstruction, and partial relaxation in the film may be at play as well [8, 45]. This warrants further study to disentangle individual contributions to the enhancement of T_c as well as to the magnetic excitations [45]. We notice that freestanding superconducting infinite-layer nickelate membranes have recently been reported [46, 47], which offer opportunities for achieving even larger strain variation without the complexity from the sample quality or interface between film and substrates.**”

we have added in the supplementary Note 1: “**It is noticeable that some broad peak weight exists in the reciprocal space maps (RSM) of the (103) reflection in Fig. S1f, indicating the film is partly relaxed.**”

we have removed the word **dramatically** and **striking** in the abstract: “the superconducting transition temperature can be **dramatically** enhanced via only $\sim 1\%$ compressive strain-tuning ... to strain-tuning, in sharp contrast to the **striking enhancement** ... ”

we have removed the word **dramatically** in the first paragraph on page 4: “...**dramatically** enhancement of T_c for superconducting films...”

Reply to Reviewer #3

We would like to thank the Reviewer for the careful review of our manuscript, and for providing constructive comments and suggestions for us to improve the manuscript. We are grateful for the Reviewer’s recognition of our efforts by “**I recognize that the authors have put some work to improve the manuscript, especially by performing new experiments and adding supporting data.**”

Below we address the questions raised by the Reviewer with more details. We indicate in **blue** the original reports, in **black** our reply, and in **red** our changes in the revised

manuscript.

(1) Reviewer’s question: “This is my second review of the manuscript entitled ”Magnetic Excitations in Strained Infinite-layer Nickelate PrNiO₂ Films” which has now been transferred to Nature Communications. I thank the authors for their detailed response and I recognize that the authors have put some work to improve the manuscript, especially by performing new experiments and adding supporting data.

The authors have addressed some of my previous comments in their response and provided corresponding modifications in the manuscript. However, I still have concerns that are related to points (4) and (5) of my previous review, that prevent my recommendation for publication of this manuscript in Nature Communications in its present form. I detail below my concerns:

To me, the raw data that clearly support the authors’ conclusions are the measurements presented on Fig. 3 c-f, that are the RIXS spectra recorded with pi-polarization on both PrNiO₂/STO vs PrNiO₂/LSAT. In these data, the magnetic excitation is clearly separated from the elastic peak for both samples and the data analysis seem robust. However, these concern only two Q points (0.48, 0) and (0.25, 0.25) and therefore this is not sufficient to determine a dispersion. The data measured with sigma-polarization on PrNiO₂/LSAT (Fig. 2 c-d) that serve to determine the dispersion plotted on Fig. 4 a, on the other hand, show a weak magnetic excitation signal that is more a shoulder of the elastic peak and phonon contributions than a clear peak. I am still troubled by the weakness of the magnetic peak signal measured with sigma-polarization on PrNiO₂/LSAT (Fig. 2 c-d) compared to PrNiO₂-STO (Fig. 2 a-b). The authors did not give a clear explanation as to why the magnetic peak is weak in these measurements. This is not the case for the measurements using pi-polarization, where both measurements on the two different samples show similar intensity for the magnon (Fig. 3 c-f). The authors should explain why the magnetic peaks in PrNiO₂-LSAT are weak in comparison to the PrNiO₂-STO sample on the sigma-polarization data (Fig. 2).

Our response: We thank the Reviewer for the excellent questions. To address the Reviewer’s concern, we have shown in Fig. R1b an additional data set obtained using π polarized incident photons for the PrNiO₂/LSAT along with those for the PrNiO₂/STO, which were from the reference [P. Worm et. al., arXiv:2312.08260] and obtained on our sample. We have performed a similar analysis as in Fig. R1a to these data sets. The solid curves represent the magnon dispersion, suggesting the magnon bandwidth for the film on LSAT is slightly reduced compared with that for the film on STO. This is consistent with the observation shown in Fig. R1a from the measurements using σ polarized incident photons. We have included this new analysis in the revised manuscript. Besides, we have amended in Supplementary Note 4 the low energy RIXS spectra as well as the fitting, which are summarized in Fig. R2.

We thank the Reviewer for pointing out that the magnetic peaks are weaker from the measurements on PrNiO₂/LSAT compared with those on PrNiO₂/STO when using σ polarized incident photons. The most likely explanation for this experimental evidence is the presence of La. Indeed the La-M₄ appears as a strong x-ray absorption signal (Fig. 1d) which in RIXS appears as elastic peak (the emission of La is at much lower energies). To reduce the contamination in the σ measurements of the elastic line we tested small different incident energies and observed that the best ratio between the elastic line and the magnon was at

Fig. R1: a,b, The mode energy (ϵ_q) and damping factor (γ_q) in PrNiO₂ grown on STO and LSAT obtained from fitting the RIXS spectra of PrNiO₂ grown on STO (LSAT) measured by σ polarization (a) and π polarization (b); the data set for the PrNiO₂ grown on STO in (b) are from reference [P. Worm et. al., arXiv:2312.08260]. The solid lines represent the best fit for the model of the spin 1/2 Heisenberg antiferromagnet on the square lattice using linear spin-wave theory. J_1 and J_2 represent the nearest-neighbour (NN) and next-nearest-neighbour (NNN), respectively. Error bars are estimated from the standard deviation obtained by the least-squares fitting algorithm and multiple times of fittings. c, Illustration of possible structural distortions in the PrNiO₂ films. d, Schematic plot of the bond-stretching phonon modes that may suppress the superexchange coupling strength.

energies 100 meV higher than the maximum of the Ni XAS. This choice, while optimizing the ratio of elastic vs magnon, decreases the overall spectral weight of the spin excitations. This is possibly the reason why the spin excitations on LSAT appear weaker compared to STO. This problem is decreased using the π polarization where the elastic is suppressed by $\cos^2(2\theta)$. However we are confident that based on the new data set added to the manuscript the Reviewer will consider this a minor detail.

Changes made: We have replaced Figure 4 by Fig. R1 in the revised manuscript. On pages 7 and 8, we have added a discussion: “Fig. 4b presents an additional data set obtained using π (linear horizontal) polarized incident photons, where the ones for PrNiO₂/STO are from the reference [35]. Similar analysis using linear spin wave theory yields J_1 (J_2) = 70.0(-8.0) meV for PrNiO₂/STO, and J_1 (J_2) = 65(-6.5) meV for PrNiO₂/LSAT, suggesting the magnon bandwidth is slightly reduced in response to the in-plane compressive strain of $\sim 1\%$ forced by LSAT referring to STO. This is consistent with the result of the measurements using σ polarized incident photons.”

We have included in Supplementary Note 4 the RIXS spectra obtained using π polarized incident photons: “In Fig. S5, we show the low-energy loss RIXS spectra that are collected using π (linear horizontal) polarized incident photons. We have performed a similar fitting procedure to that employed in Fig. S4 to resolve individual components in the RIXS spectra. The magnon energies and damping factors for each momentum position obtained from the fitting are shown in Fig. 4b.

(2) Reviewer’s question: “I am also still troubled by the extracted position of the mag-

Fig. R2: RIXS spectra of the PrNiO₂ film grown on LSAT along (h, 0) and (h, h) directions. The filled black circles represent the data and the solid red curves fit the data set, using a combination of an elastic scattering contribution (green), a Gaussian profile for the phonon peak (orange), a DHO function for the magnetic excitation (purple), and background (grey). All the measurements were taken at 16 K with π polarized incident photons.

netic excitations on both PrNiO₂-LSAT and PrNiO₂-STO samples that changes substantially between measurements with sigma-polarization and those with pi-polarization, i.e. red circles and blue squares vs red and green diamonds on Fig. 4 a (almost 25 meV difference for PrNiO₂-STO). While I understand that at the end the results are equivalent between the two pi and sigma measurements, I however wonder if the authors have an explanation for such difference between pi and sigma measurements? As the authors stated in their response, the spectra can be different between sigma and pi (“larger contrast between elastic / phonon and magnetic excitations, and weaker spectra distorted by self-absorption”), but are not supposed to give different positions for the extracted peaks. Does this difference affect the error determination of the magnetic peak position and estimation of the super-exchange coupling J?

Our response: We thank the Reviewer for pointing this out. We extract the magnetic peak positions by fitting the RIXS spectra with multiple components. In the case of using σ polarized incident photons, overlapping between the phonon and magnon peaks in the current energy resolution of RIXS should result in some uncertainties in determining the peak positions at low in-plane momentum, which however should lead to systematic error rather than random error. This is also the reason why we only compare the change in the magnon bandwidth for the same photon polarization and the same scattering geometry, which eventually yield consistent results as shown in Fig. R1 in both cases. As a note the self absorption can induce a small change in the energy because the intensity correction scales differently based on the energy loss. This means that a strongly self-absorbed peak

Fig. R3: RIXS spectra of the PrNiO₂ film grown on LSAT along (h, 0) and (h, h) directions. The filled black circles represent the data and the solid red curves fit the data set, using a combination of an elastic scattering contribution (green), a Gaussian profile for the phonon peak (orange), a DHO function for the magnetic excitation (purple), and background (grey). All the measurements were taken at 40 K with σ polarized incident photons.

will appear at higher energies than a weakly self-absorbed one.

(3) Reviewer’s question: “Considering the data quality shown on Fig. 2 c-d, I suggest to plot these data and fit contributions with a zoom on the interesting part, i.e. plotting with a smaller intensity scale for instance or by changing the aspect ratio in order to better highlight the spectra and corresponding fits near the magnetic peak, or at least to show such zoom in the Supplementary Information. With the current plotting format, the reader cannot

judge the quality of the fit, especially on the tail of the magnetic peak for the PrNiO₂/LSAT sample.

Our response: We thank the Reviewer for the excellent suggestions. We have added in Supplementary Note 4 an enlarged view of Fig. 2 c-d for the reader to judge the quality of the fit (see Fig. R3 in our reply).

Changes made: We have added in Supplementary Note 4 an enlarged view of Fig. 2 c-d for the reader to judge the quality of the fit: “In Fig. S4, we show an enlarged view of the low-energy loss RIXS spectra of Fig. 2c-d in the main text. This allows better visualization of the agreement of the fitting to the raw data. Overall, the fit shows a good agreement to the data enabling us to resolve individual components properly.”

(4) Reviewer’s question: “Again, considering the data quality shown on Fig. 2 c-d, I still have concerns about the fitting model of the PrNiO₂/LSAT RIXS spectra and therefore wonder how accurate are the extracted magnetic peak positions along with the corresponding error bars plotted on Figure 4 (a). What are the results of the fits on the PrNiO₂/LSAT sample (Fig. 2 c-d) if the magnon peak position is fixed to the expected value to scale with an increase of T_c by 40%? The authors should perform this counter analysis and compare to the current model.”

Our response: We fully understand the Reviewer’s concerns about the fitting model of the PrNiO₂/LSAT RIXS spectra. Following the Reviewer’s excellent suggestions, we have analyzed the tolerance in the fitting by fixing the magnon peak position to scale with an increase of 40% in T_c . Near the Brillouin zone boundary $(h, k) = (0.5, 0)$ and $(0.25, 0.25)$, the magnon peak position is approximately proportional to the magnon bandwidth. Figures R4a and b show the low energy loss RIXS spectra $(0.47, 0)$ and $(0.25, 0.25)$ collected using σ polarized incident photons, respectively. The fit is given by the solid red curve, which shows a good agreement to the data. In Fig. R4c and d, we fix the magnon peak positions with an enhancement of 40%. This yields a substantial deviation between the fit (solid red curves) and the data. We have performed a similar analysis for the data measured using π polarized incident photons. As shown in Figures R5 c and d, the solid red curves represent the fit to the data with the magnon peak positions fixed to scale with a 40% increase to the corresponding values shown in R5 a and b; likewise, they deviate substantially from the measurements.

Changes made: We have added in Supplementary Note 6 the counter analysis as suggested by the Reviewer: “As shown in Fig.4a, the magnon bandwidth for PrNiO₂/LSAT is slightly smaller than that for PrNiO₂/STO. This is further confirmed in the measurement using different scattering geometry with π polarized incident photons, as summarized in Fig. 4b. In the case of the σ and grazing incidence geometry, the data shown in Fig. 2c-d (enlarged in Fig. S4) appear not as good as those for PrNiO₂/STO. In the fitting procedure, multiple components were used in the fitting to achieve a good agreement to data as shown in Fig. S4. Nevertheless, it is still necessary to assess the tolerance in the fitting by fixing the magnon peak position to scale with an increase of 40% in T_c . Near the Brillouin zone boundary $(h, k) = (0.5, 0)$ and $(0.25, 0.25)$, the magnon peak position is approximately proportional to the magnon bandwidth. Fig. S6a and b show respectively the low energy loss RIXS spectra at $(0.47, 0)$ and $(0.25, 0.25)$ obtained using σ polarized incident photons. The fit is given by the solid red curve, which shows a good agreement to the data. In Fig. S6c and d, we fix

Fig. R4: Fitting the RIXS spectra with the magnon position fixed to scale with an increase of 40% in Tc. a and b show respectively the RIXS spectra at (0.47, 0) and (0.25, 0.25). The filled black circles represent the data and the solid red curves fit the data set. c and d represent the same data set corresponding to a and b, but the solid curves are the fits with magnon peak position fixed.

the magnon peak positions with an enhancement of 40%. This yields a substantial deviation between the fit (solid red curves) and the data. We have performed a similar analysis for the data measured using π polarized incident photons. As shown in Fig. S7c and d, the solid red curves represent the fit to the data with the magnon peak positions fixed to scale with a 40% increase to the corresponding values shown in Fig. S7a and b; likewise, they deviate substantially from the measurements.”

Summary of Changes

Below, we summarize the changes made in the revised manuscript.

1. On page 2, we have removed the word **dramatically** and **striking** in the abstract: “the superconducting transition temperature can be **dramatically** enhanced via only $\sim 1\%$

Fig. R5: Fitting the RIXS spectra with the magnon position fixed to scale with an increase of 40% in T_c . a and b show respectively the RIXS spectra at $(0.48, 0)$ and $(0.25, 0.25)$. The filled black circles represent the data and the solid red curves fit the data set. c and d represent the same data set corresponding to a and b, but the solid curves are the fits with magnon peak position fixed.

compressive strain-tuning ... to strain-tuning, in sharp contrast to the **striking enhancement ...** ”

2. On page 3, we have changed the sentence “Such an enhancement of T_c is connected with strain.” to “**therefore calling for a connection with strain**”

3, on pages 7 and 8, we have added a discussion: “**Fig. 4b presents an additional data set obtained using π (linear horizontal) polarized incident photons, where the ones for PrNiO₂/STO are from the reference [35]. Similar analysis using linear spin wave theory yields $J_1 (J_2) = 70.0(-8.0)$ meV for PrNiO₂/STO, and $J_1 (J_2) = 65(-6.5)$ meV for PrNiO₂/LSAT, suggesting the magnon bandwidth is slightly reduced in response to the in-plane compressive strain of $\sim 1\%$ forced by LSAT referring to STO. This is consistent with the result of the measurements using σ polarized incident photons.**”

3. On page 9, we have used a rather measured tone on the enhancement of T_c versus strain

considering contrasting results across the groups, and have further softened our conclusion in the revised manuscript by emphasizing: “Admittedly, it is possible that besides the electronic modification via strain, other effects such as crystalline quality, interface reconstruction, and partial relaxation in the film may be at play as well [8, 45]. This warrants further study to disentangle individual contributions to the enhancement of T_c as well as to the magnetic excitations [45]. We notice that freestanding superconducting infinite-layer nickelate membranes have recently been reported [46, 47], which offer opportunities for achieving even larger strain variation without the complexity from the sample quality or interface between film and substrates.”

4. On page 20, we have replaced Fig.4 with Fig. R1.

5. We have added in the supplementary Note 1: “It is noticeable that some broad spectra weight exists in the reciprocal space maps (RSM) of the (103) reflection in Fig. S1f, indicating the film is partly relaxed.”

6. We have included in Supplementary Note 4 the RIXS spectra obtained using π polarized incident photons: “In Fig. S5, we show the low-energy loss RIXS spectra that are collected using π (linear horizontal) polarized incident photons. We have performed a similar fitting procedure to that employed in Fig. S4 to resolve individual components in the RIXS spectra. The magnon energies and damping factors for each momentum position obtained from the fitting are shown in Fig. 4b.

7. We have added in Supplementary Note 4 an enlarged view of Fig. 2 c-d for the reader to judge the quality of the fit: “In Fig. S4, we show an enlarged view of the low-energy loss RIXS spectra of Fig. 2c-d in the main text. This allows better visualization of the agreement of the fitting to the raw data. Overall, the fit shows a good agreement to the data enabling us to resolve individual components properly.”

8. We have added in Supplementary Note 6 the counter analysis as suggested by the Reviewer: “As shown in Fig.4a, the magnon bandwidth for PrNiO₂/LSAT is slightly smaller than that for PrNiO₂/STO. This is further confirmed in the measurement using different scattering geometry with π polarized incident photons, as summarized in Fig. 4b. In the case of the σ and grazing incidence geometry, the data shown in Fig. 2c-d (enlarged in Fig. S4) appear not as good as those for PrNiO₂/STO. In the fitting procedure, multiple components were used in the fitting to achieve a good agreement to data as shown in Fig. S4. Nevertheless, it is still necessary to assess the tolerance in the fitting by fixing the magnon peak position to scale with an increase of 40% in T_c . Near the Brillouin zone boundary $(h, k) = (0.5, 0)$ and $(0.25, 0.25)$, the magnon peak position is approximately proportional to the magnon bandwidth. Fig. S6a and b show respectively the low energy loss RIXS spectra at $(0.47, 0)$ and $(0.25, 0.25)$ obtained using σ polarized incident photons. The fit is given by the solid red curve, which shows a good agreement to the data. In Fig. S6c and d, we fix the magnon peak positions with an enhancement of 40%. This yields a substantial deviation between the fit (solid red curves) and the data. We have performed a similar analysis for the data measured using π polarized incident photons. As shown in Fig. S7c and d, the solid red curves represent the fit to the data with the magnon peak positions fixed to scale with a 40% increase to the corresponding values shown in Fig. S7a and b; likewise, they deviate substantially from the measurements. ”

Reviewers' Comments:

Reviewer #2:

Remarks to the Author:

I thank the authors for their detailed answer. I have gone through the reply letter and the revised manuscript. The answers are convincing and the revised manuscript addresses all of my concerns. I really appreciate that the authors added new data to support their claim along with performing the proposed counter analysis.

This study and these data should be published, the results being of great interest for the RIXS and Nickelate community.

Reply to Reviewer #2

Reviewer 2 (Remarks to the Author):

I thank the authors for their detailed answer. I have gone through the reply letter and the revised manuscript. The answers are convincing and the revised manuscript addresses all of my concerns. I really appreciate that the authors added new data to support their claim along with performing the proposed counter analysis.

This study and these data should be published, the results being of great interest for the RIXS and Nickelate community. ”

Our response: We thank the reviewer for thoroughly assessing our reply to the reviewers' comments and the revised manuscript. We are very grateful to the reviewer for the critical appraisal of our work and for suggesting its publication in Nature Communications.